# When Nurses Become Patients. Validation of the Content of the Diagnostic Label Professional Traumatic Grief

**DOI:** 10.3390/healthcare9081082

**Published:** 2021-08-23

**Authors:** Ester Gilart, Isabel Lepiani, María José Cantizano Núñez, Inmaculada Cabrera Roman, Anna Bocchino

**Affiliations:** 1Doctor López Cano Hospital of Cadiz, 11010 Cadiz, Spain; esther.gilart@gmail.com; 2Nursing University Salus Infirmorum of Cadiz, 11001 Cadiz, Spain; Isabel.lepiani@ca.uca.es; 3Emergency Department Bahia de Cadiz, La Janda, 11001 Cadiz, Spain; mjcanty14@hotmail.com (M.J.C.N.); valentinaspain@hotmail.com (I.C.R.)

**Keywords:** nursing, traumatic grief, Covid-19

## Abstract

(1) Background: The Covid-19 pandemic has put our healthcare systems to the test, exposing their vulnerability and demanding a high degree of commitment and dedication from healthcare teams to confront and contain the disease. In this sense, nursing professionals have played a prominent role in the treatment of complex cases that have required hospitalisation and have manifested pre-existing health complications or mortality. This unprecedented situation made it difficult to regulate the emotional impact and manage grief, often turning it into a traumatic grief whose psychological and emotional manifestations are increasingly evident but very little researched in the current context. (2) Purpose: Validation of the definition, defining characteristics and related factors for the proposal of the nursing diagnosis of professional traumatic grief. (3) Method: Based on Fehring’s content validation model, the label name, defining characteristics and other related factors were agreed upon by Spanish experts. (4) Results: The content validity index was 0.9068. A total of 21 defining characteristics were validated by the experts, as all of them scored above 0.6. With respect to the related factors of the proposed 10, all were validated. (5) Conclusion: The present study supports the proposal to develop a nursing diagnosis for professional traumatic grief. The use of standardised language is only the first step in establishing professional traumatic grief as a diagnostic category.

## 1. Introduction

In the healthcare context, grieving events are quite frequent, and even though death is considered an inevitable part of the individual’s life process, healthcare professionals are not trained to deal with such situations or to deal with the emotions and feelings that arise during a grieving process [1,2,3,4]. When a patient dies, be it a child, an adult or an elderly person, the health professional has to face the bereavement by dealing with all the emotions that this may entail, since he/she has to provide the best possible care to the relatives of the deceased, accompanying them in the first delicate phases of the post-traumatic shock [5]. These professionals are not exempt from the emotional influence of death, going through a grieving process themselves and a variety of emotions and feelings that can affect their health. The healthcare professional is capable of feeling pain and suffering, to a greater or lesser extent, when a patient dies [6]. Facing the death of a patient, which can sometimes be considered as something unnatural and incomprehensible, has an impact on the professional’s biopsychosocial state and work performance [7,8]. Research with nursing professionals has shown that this population presents alterations in their physical, mental and social health, as a consequence of their experiences when attending the death of patients [9]. 

Nurses, for example, present contrasting emotions, ranging from nervousness, helplessness, uncertainty, guilt and frustration to anger, sadness and others [10]. Even in critical care units, where a higher number of deaths occur, nurses express multiple emotions [11]. Most of the time these emotions are resolved in a natural way, in terms of their painful reactivity and duration. However, there are other circumstances in which they do not follow the expected course, causing alterations in the individual’s normal functioning and affecting his or her quality of life, which is another form of bereavement, complicated or pathological [12]. Although there is no consensus on the criteria and designations of complicated grief (e.g., pathological, complicated, traumatic, protracted, chronic or morbid grief), its inclusion in the different diagnostic manuals currently assumes the following designations: “Persistent Complex Grief-Related Disorder” (PCB-RD) for the 5th edition of the Diagnostic and Statistical Manual of Mental Disorders (DSM-5) [13] and “Prolonged Grief Disorder” (PGD) for the 11th edition of the International Classification of Diseases (ICD-11) [14]. These new designations provided valuable insight and a solid foundation that distinguishes, or at least attempts to differentiate, complicated grief from other types of disorders such as depression or post-traumatic stress disorder. Thus, the push to have prolonged grief included in diagnostic systems has reignited the debate on the topic of pathological grief, with various therapists and researchers raising objections and concluding that it is a diagnosis that requires further study [15,16]. However, from a nursing perspective, the diagnostic labels “complicated grieving” and “risk of complicated grieving” also included in the latest update of the taxonomy for International Nursing Diagnoses (NANDA) [17] do not bring significant changes from previous versions of the manual. Therefore, if recent research [18] has highlighted the importance of treating and offering therapeutic alternatives to people suffering from possible pathological persistent grief, the inclusion or updating of complicated grief by healthcare professionals would be a good starting point to optimise comprehensive and holistic care. Finally, knowing the predictors of complicated grief could facilitate the prevention, follow-up and treatment of these patients, as it would allow a better identification of the possible symptomatology to be treated. Numerous authors [18,19,20,21,22] have detailed how personal factors, those related to the deceased, those related to the illness or circumstances of the death and those related to relational aspects play an important role in the deterioration after a loss. Social support has been one of the variables that has received most attention regarding this topic.

Research indicates that social support from family and close friends is essential at the time of bereavement to facilitate the process and to rebuild social relationships. A small but growing body of knowledge demonstrates the important role of social support in reducing the impact of sudden loss on depressive and bereavement symptomatology [23].

Another aspect of particular relevance, especially in the current context, dictated by the Covid-19 pandemic, are the factors related to the illness and the circumstances of death. Indeed, sudden deaths, painful disease processes and uncontrolled symptoms, mistrust or doubts about the medical treatment followed, are considered predictors of pathological bereavement. Even other factors, such as multiple deaths or obligations, are considered as such. The type and volume of losses a person experiences also affect the grieving process and the likelihood of prolonged grief. Researchers have predicted that the circumstances and characteristics of Covid-19-related deaths will lead to a worldwide increase in persistent and disabling grief reactions [24,25]

As a result, the situation has become more complicated over the past two years. The situation becomes more complicated, not only for those individuals or families who have not been able to assist the person in their terminal state, but also for the health professionals themselves who have had to face an unprecedented situation [26].

The Covid-19 pandemic has put our healthcare systems to the test, exposing their vulnerability and demanding a high degree of commitment and dedication from healthcare teams to confront and contain the disease. In this regard, nursing professionals have played a leading role in the treatment of complex cases that have required hospitalisation and have manifested pre-existing health complications or mortality [27].

This situation has brought with it a series of bio-psycho-social repercussions on health professionals, particularly nurses, who are the ones who provide personalised care and comprehensive attention to hospitalised patients on a daily basis. In the same context, what would happen if the same health professionals became people in need of treatment or therapy because they did not know how to cope with a complicated bereavement? The impossibility of anticipating the events, the multiple sudden deaths of infected patients in a short period of time, the excessive workload of the health team and the lack of health personnel and the necessary means of protection made it difficult to regulate the emotional impact and the elaboration of grief until it became, on many occasions, a traumatic grief whose psychological and emotional manifestations become more and more evident [8,28].

These include anxiety, fear and other emotional states, such as feelings of helplessness, insomnia, psychological distress, exhaustion, depressive symptoms, somatisation, and feelings of stigmatisation and frustration [29,30,31].

Some manifestations of post-traumatic stress disorder (PTSD) have even been observed in these health professionals [32].

If we add to all this the fact that these consequences can impair the mood of these workers, which translates into lower work performance and a greater risk of involuntary errors or omissions, the situation becomes urgent and paramount.

Despite the clinical importance of the topic, no quantitative research has comprehensively analysed the bereavement experienced by health professionals due to the pandemic, let alone the health consequences of bereavement due to Covid-19 [33]. This situation in itself requires an evidence-based response provided by the scientific community, while providing quality, comprehensive and systematised care [34], in this case for the same nurses. Covid-19 has ceased to be considered an abstract phenomenon and has become an individual, family and generalised reality that affects a multitude of people, professions and sectors [35]. Therefore, and according to the studies reviewed, interventions should be aimed at strengthening the threat to personal identity, fostering coping resources in healthcare professionals, helping to minimise the negative self-evaluation of grief, and improving the quality of life and healthcare of at-risk populations.

In this regard, the role of health professionals is essential to addressing this problem since, in Spain alone, 80,579 have died in the last year [36]. It is necessary to carry out a joint assessment of all aspects, both physical and psychological, of the individual, and to ensure that this is applied within the main areas of intervention (clinical, labour, school, sports, etc.) with the provision of the corresponding health services. Nursing professionals can provide other health professionals and ourselves with strategies to cope with such situations and, if they are difficult to cope with, provide the appropriate knowledge and skills to manage the negative experiences or consequences that the current situation has produced. Nursing professionals have to be prepared and trained to face this situation knowing the available resources, and thus being able to overcome the limitations and reach higher levels of health and well-being.

The present study proposes a different approach to bereavement, “Professional traumatic grief” adapted not only to the current context but also to the grief experienced by health professionals in certain circumstances and within their work context. It is necessary to analyse the symptoms and related factors of bereavement from a nursing perspective, identifying the needs that professionals face in the new work context. This study is part of a broader investigation into the bereavement experienced by nurses in a new situation where multiple unexpected deaths have been witnessed. In addition, the use of standardized scientific language may be the catalyst that enables the approach from theory to practice and, therefore, the necessary feedback to provide systematic high-quality care based on evidence.

In this vein, the main objective of this study is the importance of inclusion and the content validation of a new diagnostic label for use in clinical and healthcare practice.

The specific objectives are as follows:The content validation of a new diagnostic label;To determine the degree of representativeness of each of the defining characteristics (DCs) of the proposed label;To determine the degree of representativeness of the other factors related (RFs) to the proposed label to evaluate the need for inclusion of the diagnosis in the NANDA-I taxonomy II.

## 2. Materials and Methods

Based on the diagnostic content validation model described by Fehring [37] and the method proposed by Walker and Avant [38], the validation of the diagnostic label should be agreed upon by a group of experts based on its definition and the content of its defining characteristics (DCs) and related factors (RFs).

Type of study, population and sample.

An exploratory, descriptive and diagnostic content validation study was conducted.

The sample chosen for convenience was made up of nurses who met the requirements to be considered experts in the nursing and care field primary care, hospitalisation and applied psychosocial sciences (teaching).

The selection was made among experts with a common and homogeneous profile in order to reach a consensus on the diagnostic label, as well as on the content of its manifestations and related factors. In order to be considered as an expert, criteria of training and healthcare experience will be taken into account. Inclusion criteria: Holding a Diploma or Degree in Nursing, minimum two years of experience in primary care or hospitalisation, minimum two years of teaching experience in the area of nursing or psychosocial sciences. Exclusion criterion: refusal to participate in the study.

Although, originally, the number of experts needed to perform diagnostic content validation was established to be 25–50 by the author of the method [39], Nunnally and Berstein [40] later determined that with 200 experts, greater stability in the analysis is achieved. Taking the judgment of these authors, a minimum sample size of 200 experts was established.

Origin and recruitment of the sample. For the recruitment of the study sample, the panel of experts was contacted by telephone and/or e-mail with collaborators of the research group requesting collaboration for the recruitment of participants. In addition, information was collected through a structured online survey, using the SurveyMonkey tool, and was sent through three main mechanisms: (a) social networks such as Facebook and Twitter, (b) institutional or academic websites (e.g., college of nursing), and (c) invitations to newsgroups and mailing lists.

The voluntary participants followed a link to the survey where they were presented with a participant information sheet and a consent form. If they agreed to continue with the survey, they automatically signed the informed consent form.

### Instruments and Variables

In addition to the definition of the diagnostic label and the DCs and RFs, sociodemographic data were collected from the experts in order to control the profile of the experts [41]. 

For the validation of the content of the Diagnostic Label based on the proposed definition, the experts in the study completed a structured survey, which was divided into the following sections: (a)Sociodemographic variables. To determine the demographic data of the experts, a questionnaire was used to collect personal data including the variables sex (Male, Female), age, academic degree (ATS/Diplomatura/Degree in Nursing, Specialty, University Expert/Non-official Master’s Degree, Official Master’s Degree, DEA/Research Proficiency, Doctorate), position in the institution in which they work (Nurse, Specialist Nurse, Case Manager, Coordinator, Teacher), and time of experience in the position (years).(b)Diagnostic label designation from definition to be named by the panel experts according to the options provided to them. The definition agreed upon by the research team was: “State of physical, psychological and social discomfort experienced by health professionals and caused by prolonged exposure to multiple sudden deaths in the work context.” The designations for the diagnostic label were: job distress, professional exhaustion, professional traumatic grief and professional shock.(c)Defining characteristics of the diagnosis. Participants were provided with a list of 33 DCs identified as the most indicative ones detected in the literature [42,43,44,45], corresponding to: fear, helplessness, shortness of breath, psychomotor agitation, guilt, anger, stress, sadness, rage, irritability, panic attacks, disturbed sleep pattern, anxiety, dizziness, feeling of unreality, state of confusion, anguish, emotional exhaustion, depression, somatisation, flashbacks, depersonalisation (clinical and listless attitude towards other patients or people), feeling of dullness, hopelessness, intrusive thoughts, tiredness, social isolation, tension, tremors, weakness, gastrointestinal disorders (stomach pain, diarrhoea, nausea, etc.), behavioural alterations (tobacco, alcohol, etc.). The experts had to identify in the chosen diagnosis those characteristics that are most representative or that can be manifested in a sufficient number of cases. For this purpose, the participants were given different values, “not at all representative”, “not very representative”, “somewhat representative”, “fairly representative” and “very representative”, to rate their degree of agreement with respect to the relationship between the defining characteristics and the diagnosis provided.(d)Diagnosis-related factors. Participants were also provided with a list of 10 RFs identified as the most representative ones detected in the literature [38,46,47,48,49,50], corresponding to: traumatic nature of the death, having witnessed sudden unexpected deaths, lack or limitation of social support, anticipatory fear (perception of more catastrophes), having witnessed multiple losses, high emotional burden, scarcity of material and human resources, work overload, high demand for care and lack of knowledge of the problem to be treated. As with the DCs, the panel of experts was asked to assess the representativeness of the proposed etiological factors.

## 3. Data Analysis

The statistical analysis was performed using the software program SPSS version 21.0 (SPSS, Chicago, IL, USA). First, the general characteristics of the sample, including socio-demographic variables, were assessed, expressing the categorical variables in terms of frequency and percentage while the mean and standard deviation were calculated for continuous variables. The diagnostic label was agreed upon in a panel of experts based on the proposed definition; the experts’ information was systematised, choosing the most frequently proposed information as the final label. For the analysis of the DCs and RFs, according to the methodology suggested by Fehring [51], the weighted mean was calculated for each DC. The weight of each DC was obtained by adding the weights assigned to each response, divided by the total number of responses. The weights assigned are: not at all representative = 0; not very representative = 0.25; somewhat representative = 0.50; fairly representative = 0.75 and very representative = 1. The proposed RFs were analysed analogously to the procedure described above. The weighted mean of the scores attributed to each of the DCs was calculated and represents the validity index diagnostic content (DCV), reaching values from 0 to 1 (0: not at all representative; 1: very representative). Depending on the scores obtained, DCs were considered highly representative or “major” (if the score was greater than or equal to 0.8); poorly representative or “minor” (if the scores are less than 0.80 but greater than 0.60) and not representative at all (if the scores are equal to or less than 0.60). The DCs that did not present significant indices were discarded. The score obtained is called the diagnostic content validity (DCV) and indicates the degree of representativeness of each DC and/or RF of the diagnostic label provided. Finally, the content validity index was obtained, which is the total score for each diagnosis, by adding the individual weights of each major or minor feature and dividing by the number of these features. 

However, in the case of diagnoses with more than seven DCs, it is recommended to calculate the overall score of the DCV index following the modifications made by Sparks and Lien-Gieschen [52]. This method consists of counting the score of the “major” features twice and that of the “minor” features once, and determining the mean of all scores. If the score obtained is greater than 0.8, the proposed diagnosis can be considered as representative. If the final score is between 0.6 and 0.79, it is difficult to establish the degree of representativeness and further study is considered necessary.

## 4. Results

The final sample consisted of a total of 210 experts, of whom 75.71% were nurses, 11.90% were specialist nurses, 1.43% were case managers, 6.67% were coordinators and 4.29% were teachers. The majority of participants were female (78.10%) with an average age of 42.7 (SD = 11.1) years. With respect to academic training, 70.48% had an ATS/Diplomatura/Degree in Nursing, 14.76% a Specialty, 5.24% a University Expert/Non-official Master’s Degree, 5.24% an official Master’s Degree, 0.48% a DEA/Research Proficiency and 3.81% a Doctorate. The average time of experience in the position was 15.2 (SD =11.3) years, 13.67% in primary care, 84.70% in hospitalisation and 1.64% in teaching. The above results are presented in detail in Table 1.

The names proposed by experts for the diagnostic label underwent categorisation through an analysis of the “units of context”. Of the four options provided, the label chosen by consensus among the experts was “Professional traumatic grief”, with a total percentage of 68.16%. The label least preferred by experts was “Professional shock”, which was proposed by only 2.17%, followed by “Professional exhaustion” (5.81%) and “Job distress” (23.86%).

With respect to the defining characteristics, of the 33 DCs proposed for the diagnosis, 21 were validated by the experts, since all had a score higher than 0.6. Critical defining characteristics were “sadness”, “stress”, “helplessness” and “fear”. With respect to the minor characteristics, “emotional exhaustion”, “sleep pattern disturbance”, “fatigue”, “anxiety”, “guilt”, “anger”, “rage”, “hopelessness”, “tension”, “anguish”, “irritability”, “gastrointestinal disturbances”, “behavioural disturbances”, “depression”, “feelings of dullness” and “flashbacks” were found. The defining features “social isolation”, “somatisation”, “shortness of breath”, “intrusive thoughts”, “panic attacks”, “psychomotor agitation”, “depersonalisation”, “sense of unreality”, “weakness”, “confusion”, “tremors” and “vertigo” were disregarded. The overall content validity index for the diagnostic label was 0.9068. The DCV index achieved by each CD is detailed in Table 2. 

With respect to the related factors of the 10 proposed FRs, all were validated, with “having witnessed multiple losses” scoring the highest (0.8178), followed by “high demand for care” (0.7936); “work overload” (0.7922); “having witnessed sudden unexpected deaths” (0.7897); “high emotional burden” (0.7657); “scarcity of material and human resources” (0.7202); “lack of knowledge of the problem to be treated” (0.7199); “traumatic nature of the death” (0.7095); “lack or limitation of social support” (0.6750) and lastly “anticipatory fear” (0.6279). Table 3 shows the results obtained.

## 5. Discussion

The main objective of the present study was to validate the content of a new diagnostic label whose definition was: “State of physical, psychological and social discomfort experienced by health professionals and caused by prolonged exposure to multiple sudden deaths in the work context”. 

Through a panel of experts, the label selected with the highest score (68.16%) was “Professional Traumatic Grief”. The scores obtained with respect to the consensus on the overall content validity index obtained for the DCs were above 0.80, indicating the conceptual validity of the work in its early stages.

Regarding the representativeness of the DCs considered by the experts as indicative for the diagnosis, the results obtained indicated that of the 33 proposed DCs, 21 satisfactorily cover the spectrum of content validation of the diagnosis evaluated. 

The representativeness of the defining characteristics such as “sadness” (0.9429), “stress” (0.8901) and “emotional exhaustion” (0.7936) are consistent with the findings of other studies whose results showed that healthcare professionals are not exempt from the emotional influence generated by the death and care of terminally ill patients, going through a grieving process themselves and a variety of emotions and feelings such as those described, which even alter their functionality [7,44,45,53].

We also highlight, for example, the high occurrence of the DCs “impotence” (0.8737) and “fear” (0.8022), and to a lesser extent “sleep pattern disturbance” (0.7895), “fatigue” (0.7809), “anxiety” (0.7573), “guilt” (0.7514), “depression” (0.6265), “hopelessness” (0.7141) and “anguish” (0.6921). This evidence seems to be in line with numerous studies that analysed the emotional bonds that healthcare professionals establish with their patients and their families, so that these situations have an impact on a personal level. Even with some typology of patients, nursing professionals have manifested alterations in their physical and mental health [9,42,54].

With respect to the more physical and/or social characteristics such as “gastrointestinal disorders” (0.6598), “fatigue” (0.7809), “feeling of dullness” (0.6148) and/or “behavioural disorders” (0.6402), it should be noted that experiencing death is a complex process that involves biopsychosocial aspects that act in synergy, enhancing and altering the quality of life and well-being of an individual [44,55], and that is why these DC have been considered representative by the experts. Even in other research [56,57], emphasis was placed on nurses’ strategies of coping with the event of death that may be religious, cultural or personal. Additionally, in the specific context of Covid-19, there are different studies that, among the many consequences, highlight symptoms and organic pathologies such as immunodeficiencies, heart disease, diabetes, hypertension and respiratory pathologies, among others [58,59,60].

These manifestations, in turn, sharpen others, such as the characteristics of “flashbacks” (0.6116), whose result has been verified in a recent research by Lima [32] that associates grief with post-traumatic stress, and “irritability” (0.6730), “Rage” (0.7445), “anger” (0.7431), “Wrath” (0.7253) and “tension” (0.7064), emotions probably triggered from an attribution of real or subjective danger, lived during a pandemic, catastrophe or health emergency [42,61].

Finally, the DCs “social isolation” (0.5785), “somatisation” (0.5669), “difficulty breathing” (0.5291), “intrusive thinking” (0.5102), “panic attacks” (0.5102), “psychomotor agitation” (0.5058), “depersonalisation” (0.4942), “feeling of unreality” (0.4942), “weakness” (0.4753), “confusion” (0.4491), “tremors” (0.3823) and “vertigo” (0.3750) were not considered to be greater than the diagnostic label “Professional Traumatic Grief”. This result differs from the information presented in the literature, which indicates that these manifestations could be typical of traumatic grief [62,63]. However, it is likely that experts have not considered these characteristics to be directly explanatory of professional traumatic grief but perhaps relate them to other diagnoses or other situations. For example, it would be interesting to consider in subsequent studies the health area where the professional works, since some of these characteristics, such as depersonalisation, could be more present in the grief process related to paediatric patients or in intensive care [63,64].

In relation to the analysis of corroborated etiological or RFs, according to the referenced literature [9,28,46,47,48,49,50], the proposed factors “having witnessed multiple losses” (0.8178), “high demand for care” (0.7936), “work overload” (0.7922), “having witnessed sudden unexpected deaths” (0.7897), “high emotional burden” (0.7657), “scarcity of material and human resources” (0.7202), “lack of knowledge of the problem to be treated” (0.7199), “traumatic nature of the death” (0.7095), “lack or limitation of social support” (0.6750) and “anticipatory fear” (0.6279) were all considered representative by the experts.

## 6. Limitations

Despite the results, the work has important limitations. Due to the voluntary nature and availability of the sample, the different health areas of each hospital and/or health centre were not considered in the present study. The inclusion of nurses with other specialties would have made it possible to reveal other findings. Likewise, another limitation is due to the fact that the majority of the sample was comprised of the female gender. In view of this, it is recommended that future studies require an equal sample of men and women, with the intention of differentiating perspectives and clinical experiences. Finally, with respect to the diagnostic label, the idea of employing an analogue scale (ranging from 0 to 100), to improve the psychometric property of the construct, could be considered for future studies. 

## 7. Conclusions

Grief is a process of adaptation that allows us to re-establish the personal equilibrium that has been altered by the loss of a loved one. Psychological manifestations vary from person to person depending on factors such as the type of relationship, the intensity and circumstances of that relationship, the unexpectedness of the loss, etc., but it always brings with it a series of negative emotions, sometimes difficult to overcome. 

In spite of being a necessary process and compatible with the present reality, there are a series of circumstances that facilitate the transition from normal grief to pathological grief. Among them, the issue addressed in this article has tried to make visible, providing it with a scientific and standardised language, the situation experienced by many health professionals during the current Covid-19 crisis. Given the results obtained and the variability of manifestations identified, it would be important to consider actively seeking and implementing further research and other methodological alternatives to continue with clinical validation on the subject.

## 8. Implications for Clinical Practice

The use of standardised language is only the first step in establishing Professional Traumatic Grief as a diagnostic category. To confirm its clinical utility, field testing in the clinic, instrument development and epidemiological studies are needed. There are still important questions regarding the application of this definition to different cultural groups and different work groups, and it is in this regard and from a nursing perspective that we must move forward and lay the foundations for the development of its clinical utility and international applicability and the improvement of quality of care. Only understanding the professional grief that nurses have experienced in the last year may help provide guidelines to prevent or minimise the detrimental effects of this crisis on health staff where institutional support will play a fundamental role, not only in the analysis of the causes and/or the individual and collective repercussions of the workers, but also in providing the affected health team with the human and material resources necessary to overcome it. A good approach to bereavement situations within the healthcare setting could prevent further harm to both healthcare team members and the general population.

## Figures and Tables

**Table 1 healthcare-09-01082-t001:** Sociodemographic characteristics of the participants.

Variables	N	Mean (SD)	Percentage
**Sex**			
Male	46		21.90
Female	164		78.10
**Age**		42.7 (11.1)	
**Length of experience in position**		15.2 (11.3)	
**Academic background**			
Nurse	148		70.48
Nurse Practitioner	31		14.76
University Expert	11		5.24
Official Master	11		5.24
Doctoral program	1		0.48
Doctorate	8		3.81
**P** **osition**			
Nurse	159		75.71
Nurse Practitioner	25		11.90
Nurse Case Manager	3		1.43
Coordinator	14		6.67
Teacher	9		4.29
**Experience in the position where you work**			
Primary Care	52		13.67
Hospitalisation	155		84.7
Teaching	3		1.64

**Table 2 healthcare-09-01082-t002:** DCV scores for each of the defining characteristics.

Defining Characteristics	Score (DCV) *	Type of Characteristic
Sadness	0.9429 (0.11)	Major **
Stress	0.8901 (0.18)	Major
Impotence	0.8737 (0.21)	Major
Fear	0.8022 (0.25)	Major
Emotional exhaustion	0.7936 (0.23)	Minor ***
Sleep pattern disturbance	0.7895 (0.24)	Minor
Fatigue	0.7809 (0.23)	Minor
Anxiety	0.7573 (0.24)	Minor
Guilt	0.7514 (0.29)	Minor
Rage	0.7445 (0.29)	Minor
Anger	0.7431 (0.28)	Minor
Wrath	0.7253 (0.30)	Minor
Hopelessness	0.7141 (0.26)	Minor
Tension	0.7064 (0.24)	Minor
Anguish	0.6921 (0.23)	Minor
Irritability	0.6730 (0.26)	Minor
Gastrointestinal disorders	0.6598 (0.27)	Minor
Behavioural alterations	0.6402 (0.28)	Minor
Depression	0.6265 (0.27)	Minor
Feeling of dullness	0.6148 (0.27)	Minor
Flashbacks	0.6116 (0.26)	Minor
Social isolation	0.5785 (0.29)	Rejected ****
Somatisation	0.5669 (0.29)	Rejected
Difficulty breathing	0.5291 (0.28)	Rejected
Intrusive thoughts	0.5102 (0.28)	Rejected
Panic attacks	0.5102 (0.31)	Rejected
Psychomotor agitation	0.5058 (0.30)	Rejected
Depersonalisation	0.4942 (0.32)	Rejected
Feeling of unreality	0.4942 (0.32)	Rejected
Weakness	0.4753 (0.29)	Rejected
Confusion	0.4491 (0.31)	Rejected
Tremors	0.3823 (0.29)	Rejected
Vertigo	0.3750 (0.30)	Rejected

* DCV: Diagnostic content validity; ** Major is ≥ 0.80; *** minor is < 0.80 and ≥ 0.60; **** rejected is < 0.60.

**Table 3 healthcare-09-01082-t003:** DCV scores for each of the related factors.

Related Factors	Score (DCV) *	Type of Factor
Witnessing multiple losses	0.8178 (0.27)	Major **
High demand for care	0.7936 (0.22)	Minor ***
Work overload	0.7922 (0.23)	Minor
Witnessing sudden unexpected deaths	0.7897 (0.28)	Minor
High emotional charge	0.7657 (0.23)	Minor
Shortage of material and human resources	0.7202 (0.24)	Minor
Lack of knowledge of the problem to be treated	0.7199 (0.31)	Minor
Traumatic nature of death	0.7095 (0.26)	Minor
Lack of or limited social support	0.6750 (0.23)	Minor
Anticipatory request	0.6279 (0.25)	Minor

* DCV: Diagnostic content validity; ** Major is ≥ 0.80; *** minor is < 0.80 and ≥ 0.60.

## Data Availability

All data were corrected in this research. Derived data supporting the findings of this study are available from the author (A.B.) on request.

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
