# Peer review of "When Nurses Become Patients. Validation of the Content of the Diagnostic Label Professional Traumatic Grief"

_healthcare, 2021, doi:10.3390/healthcare9081082_

Round 1

Reviewer 1 Report

I find this manuscript very interesting and informative. The Professional Traumatic Grief is highly relevant in health care providers overall in the pandemic era. In addition, the authors of the manuscript show a statistical level suitable for Healthcare readers.

However, I would like to suggest some comments/modifications in order to help improve the manuscript:

There are a few sentences with awkward language use and run-on sentences, in addition to some Spanish expressions (lines 154, 181, 274, 278).

Material and Methods section should be reorganized. The titles of the subsections are repeated and the paragraphs are not correctly separated.

Should the interpretation of the VCD index be included, i.e., what does it mean to have an index value of 1?

For continuous variables in Table 1, the title "Mean (SD)" is noted in the same box that contains the value of the variable. I recommend including the title "Mean (SD)" in the table header.

Variable names appear throughout the text, both in quotation marks and not. I would suggest homogenizing the text and putting them all in quotation marks or all without quotation marks.

Limitations sections should include a separate header.

Author Response

Reviewer: 1                                                                                             

Reviewer:I find this manuscript very interesting and informative. The Professional Traumatic Grief is highly relevant in health care providers overall in the pandemic era. In addition, the authors of the manuscript show a statistical level suitable for Healthcare readers.

Response: Dear reviewer, we thank you in advance for all the contributions and sound advice given to improve the article.

Reviewer: However, I would like to suggest some comments/modifications in order to help improve the manuscript:

There are a few sentences with awkward language use and run-on sentences, in addition to some Spanish expressions (lines 154, 181, 274, 278).

Response: Dear reviewer, thank you very much for your input. The article was translated by a specialized translation agency , they were the ones who decided to use the expression. However it is true, as you suggest, there are some errors in the text. All of the above mentioned sections have been reviewed and modified. Thank you very much.

Reviewer: The Material and Methods section should be reorganized. The subsection headings are repeated and the paragraphs are not correctly separated.

Response: Dear reviewer, thank you for your input and suggestions. We have proceeded to modify the Material and Method section as suggested.

Reviewer: Should the interpretation of the VCD index be included, i.e., what does it mean to have an index value of 1?

Response: Dear reviewer, thank you very much for your suggestion. We have included this in the data analysis section (page 5).

Reviewer: In the case of continuous variables in Table 1, the title "Mean (SD)" is entered in the same box that contains the value of the variable. It is recommended to include the title "Mean (SD)" in the heading of the table.

Response: Dear reviewer, thank you for your input, we have proceeded to modify the table as you suggested (table 1, page 6).

Reviewer: The variable names appear throughout the text, both in quotation marks and without them. I suggest homogenizing the text and putting them all in quotation marks or all without quotation marks.

Response: Dear reviewer, thank you for your attention to the article. We have revised all the text as suggested. Thank you very much

Reviewer: Limitations sections should include a separate header.

Response: Thank you very much for your input. We have proceeded to separate the limitations section (Page 9). Thank you 

Reviewer 2 Report

This is a descriptive study on the measurement of the nurse traumatic grief construct.  The report illustrates how content validity or face validity could be applied in the development of refined measurement of an important construct applicable to the examination of the impact of COVID-19 on nurses' wellbeing.

Two areas could be strengthened as follows:

  1. The validation method or procedure shows that the items relevant to the "traumatic grief" could be classified into three categories.  In fact, if one employs an analog scale (ranging from 0 to 100), it would improve the psychometric property of the construct substantially.  In other words, the authors could document this limitation and present a better or refined approach in scaling.
  2.  The measurement items related to the "traumatic grief" construct should be subjected to a phi-correlation analysis so that the authors could perform a preliminary construct-validity analysis.
